# COVID-19 Vaccine Hesitancy: A Content Analysis of Nigerian YouTube Videos

**DOI:** 10.3390/vaccines11061057

**Published:** 2023-06-02

**Authors:** Mohammed Sadiq, Stephen Croucher, Debalina Dutta

**Affiliations:** 1School of Communication, Journalism and Marketing, Massey University, Wellington 6022, New Zealand; m.sadiq@massey.ac.nz; 2School of Communication, Journalism and Marketing, Massey University, Palmerston North 4442, New Zealand

**Keywords:** COVID-19 vaccine, theory of planned behaviour, health communication, content analysis

## Abstract

Vaccination is key to developing herd immunity against COVID-19; however, the attitude of Nigerians towards being vaccinated stalled at the 70% vaccination target. This study engages Theory of Planned Behaviour to analyse the tone of Nigerian YouTube headlines/titles, and the tone of YouTube users’ comments to examine the causes of COVID-19 vaccine hesitancy. YouTube videos uploaded between March 2021 and December 2022 were analysed using a content analytic approach. Results show 53.5% of the videos had a positive tone, while 40.5% were negative, and 6% neutral. Second, findings indicate most of the Nigerian YouTube users’ comments were neutral (62.6%), while 32.4%, were negative, and 5% were positive. From the antivaccine themes, analysis shows the people’s lack of trust in the government on vaccines (15.7%) and the presence of vaccine conspiracy theories mostly related to expressions of religion and biotechnology (46.08%) were the main causes of COVID-19 vaccine hesitancy in Nigeria. The study presents implications for theory and recommends ways for governments to develop better vaccination communication strategies.

## 1. Introduction

Effective communication about behavioural change related to the COVID-19 vaccination entails the ability of people to comply with health communication messages designed to create awareness, reduce concerns, inform, and counter fears related to COVID-19 vaccines and build the required willingness to be vaccinated [1]. Social and behavioural barriers, nevertheless, are influential factors that can affect how effectively COVID-19-related behavioural change is communicated [2,3]. People’s refusal to adhere to COVID-19 protocols has contributed to spikes in the number of COVID-19 infections and deaths [4]. Consequently, the World Health Organization (WHO) advised vaccination as key to developing herd immunity against COVID-19 [5]. Herd immunity occurs when a significant percentage of people (i.e., the population) develop immunity against infectious diseases through vaccination; as such, the chances of disease transmission from one person to another will considerably decrease. The potency of behavioural change communication has been challenged because of growing misinformation since the emergence of the COVID-19 vaccination rollout in 2020 globally [3,6,7,8,9,10]. Researchers discovered relevant types of information and sources related to people’s beliefs assist in shaping behavioural responses and intentions and are critical to public health crisis management and preparedness [11].

Researchers have observed that the influence of misinformation emanating from new media channels has affected and contributed to the increased causes of hesitancy against COVID-19 vaccination efforts globally [12]. Similarly, in Nigeria, the influence of COVID-19 misinformation, particularly on new media platforms, has contributed to only 33.86% of the population being fully vaccinated, even though a 70% vaccination rate was projected by the end of 2022 [13,14,15]. It is pertinent, therefore, to examine the influence of social and behavioural barriers against peoples’ intentions and willingness to receive the COVID-19 vaccination in Nigeria. This study examines the link between COVID-19 vaccination in Nigeria and YouTube, a popular new media platform in the nation. Specifically, this study has three aims. First, this study identifies the tone of YouTube video headlines/titles related to the COVID-19 vaccination campaign in Nigeria. Second, this study describes the tone of Nigerian YouTube users’ comments on Nigerian COVID-19 vaccine videos. Third, this study explains the causes of COVID-19 vaccine hesitancy identified within Nigerian YouTube video user comments.

### 1.1. Health Communication on YouTube

According to the World Health Organisation (WHO), there have been 266,463 confirmed cases of the COVID-19 pandemic in Nigeria and 3155 deaths, as of 15 February 2023. According to a recent study, several Nigerian states are vulnerable to the COVID-19 pandemic due to the high coronavirus (SARs CoV-2) seroprevalence of approximately 78.9% due to low vaccination rates [16]. Serology is the examination of blood samples tested for antibodies built against (SARS-CoV-2), the virus that causes COVID-19 [15]. YouTube accounts for 5.29 million user channels in Nigeria, a country with 33 million active new media users [17,18]. YouTube is one of the most popular new media platforms in Nigeria and is particularly relevant for sharing and receiving public health information on the COVID-19 pandemic [19,20].

Internet penetration has created the emergence of new media platforms (WhatsApp, Twitter, Facebook, TikTok, YouTube, etc.), that provide easy access to online health communication and public health information. Various social networks, like Facebook, WhatsApp, Instagram, and YouTube, are being leveraged to disseminate information about the COVID-19 pandemic [21]. Making health communication video ads has become a popular campaign communication channel (e.g., the role of microcelebrities during the COVID-19 pandemic) on YouTube [19]. A complex story can be simply understood in an eye-catching way with the help of a YouTube video campaign narrated and presented in an entertaining and informative manner [22,23]. This development explains why YouTube is influential in public service campaign advertising used to distribute promotional videos to educate people on the COVID-19 pandemic [24].

According to Mohsin [25], YouTube has a record 2.3 billion active users, over a billion hours of video watched in 100 countries, 80 languages, and 400 h of videos uploaded every day. This makes many of the health communication messages disseminated on YouTube highly recognised by users and acknowledged as the primary source of information, knowledge, and awareness creation on public health and individual well-being [26]. Nigeria has 33 million active users of new media channels like WhatsApp, Facebook, and YouTube [17]. YouTube became a major source of information in 2019 that predicted the health behaviours of social media users, on whom many people rely for information about the COVID-19 pandemic [27]. Despite its vital role in conveying important public health information, scholars have noticed that YouTube is also being used as a source of misinformation and disinformation in public health, particularly for promoting negative and false information about COVID-19 [19].

### 1.2. Misinformation on COVID-19 Vaccination Communication on YouTube

Vaccine hesitancy is not a recent occurrence in Nigeria. Research shows many people are opposed to vaccinations, especially with the backlash against polio vaccines 30 years ago [28,29,30]. However, with the COVID-19 pandemic, misinformation disseminated through social media, such as YouTube [19], has helped the spread of conspiracies that affected the COVID-19 vaccine uptake in Nigeria [31], exponentially escalating vaccine hesitancy [31,32].

The spread of misinformation during pandemics has been a documented practice since the Middle Ages [33]; however, the viral nature of misinformation on social media is a significant source of concern as well as a threat to public health [34]. Since the beginning of the COVID-19 pandemic, fake news has been widely disseminated, making people distrust the integrity of information from healthcare services and political authorities [35]. Misinformation is derived from concepts mostly related to falsehoods and fake news on new media and other information ecologies, sometimes including social forces [36]. Misinformation has created serious public panic, which has led to support for harmful public health practices against the COVID-19 pandemic [37]. Misinformation on YouTube is a major source of concern in public health crises such as the COVID-19 vaccination campaign [19], especially as the channel is used in fostering rumours and exponentially spreading conspiracy theories on COVID-19 vaccines [9,38]. According to Krishna [39], the rate at which misinformation about vaccinations spreads, in general, is worrisome enough to be classified as an epidemic. Misinformation about major pandemic outbreaks is arguably one of the main reasons why they linger, as with HIV/AIDS, or more recently with the Ebola outbreak in West Africa [40]. Unverified sources posted on new media platforms play critical roles in reporting false numbers of COVID-19 cases, providing inaccurate guidelines, and promoting unapproved therapies and remedies as antiviral cures [35,41,42,43]. Misinformation promotes social anxiety about health, giving rise to chaos, dishonesty, financial exploitation, and fear [44]. Fear and misunderstanding created by misinformation largely affect people’s willingness to report their well-being and seek treatment [45].

Krishnan and Thomson [40] concluded a major and controversial form of health misinformation is about COVID-19 vaccines. Several studies have closely examined this area of misinformation and concluded that the media influences public perceptions of vaccine disputes. Therefore, through numerous efforts to erase misconceptions about vaccines and reduce vaccine hesitancy, extant scholarship suggests a few strategies to address the issue [40,44,46]. To counter misinformation, the concept of health literacy is key to any discussion of health-related information. It refers to people’s ability to obtain, process, and understand basic health information and services necessary for making informed health decisions [47]. However, according to Krishnan [39], misinformation has created a barrier to people’s ability to attain health literacy, since those armed with incorrect knowledge would be ill-equipped to make proper health decisions. Misinformation about health issues has persisted, particularly on social media, increasing vaccine hesitancy [34,40,44] because misinformation is commonly used on the continuum to establish the scope of vaccine acceptance and vaccine rejection [33].

Misinformation has expanded because Internet penetration and networking sites like YouTube [48] have allowed users to discuss and share health-related misinformation and ideas with potentially large audiences [34,48]. Therefore, the problem for health communication research is how to evaluate the accuracy of such messages shared online and used in addressing future research directions [40]. Studies have shown sharing accurate information, particularly through YouTube, is an important component of health communication [44]. According to Mheidly and Fares [44], providing people with reliable scientific information can be useful during pandemics, especially when it is honest and accurate. In contrast, false information can exacerbate the pandemic by supporting inadequate or risky behaviours and promoting unsafe practices.

### 1.3. Social and Behavioural Barriers

The Theory of Planned Behaviour (TPB) [49,50] predicts that human behaviour is readily influenced by three concepts. First, attitude towards the behaviour indicates the degree to which a person(s) views an action as being favourable or unfavourable [51]. Second, subjective norm refers to the belief that most people who matter to an individual believe he or she should or should not engage in the behaviour in question [49,50]. Third, perceived behavioural control determins the perceived ease or difficulty of carrying out the behaviour [49,50]. In this regard, behavioural beliefs can influence how people exhibit a positive or negative attitude towards the behaviour [52] It has been established that peoples’ beliefs related to social institutions (opinion leaders and sociopolitical institutions) and behavioural-related factors characterised by trust, attitudes, knowledge, awareness, and perceived side effects of the COVID-19 vaccine are influential on the willingness or intention to be vaccinated [9,53,54].

This study argues that these dimensions serve as important guides that model the social and behavioural factors that shape YouTube users’ comments on COVID-19 vaccination hesitancy in Nigeria. The constructs, however, demonstrate that normative beliefs are related to social pressure or subjective norms, while behavioural beliefs present positive or negative attitudes towards the behaviour, and control beliefs produce perceived behavioural control, which entails the willingness or difficulty to execute the behaviour [49].

People’s personal experiences of fear, disbelief, mistrust, and anxiety are critical considerations for the intention to receive the COVID-19 vaccination. As such, understanding the fundamentals of interacting with different people, particularly those whose preconceived opinions are shaped by social-behavioural factors, is necessary for the COVID-19 vaccination process [55]. Therefore, overcoming vaccine hesitancy is related to increasing public trust and willingness to be vaccinated against the COVID-19 virus [56]. It is important to note that the COVID-19 pandemic has made many people sensitive and anxious; hence, understanding their views is critical, and communicating with them about the vaccine should take an encouraging tone that is honest, accurate, and truthful [57,58].

On the other hand, it is crucial to also discuss the risks involved in not being vaccinated and the repercussions should people refuse to do their part in achieving herd immunity through vaccination, particularly those who are dismissive or doubtful about the COVID-19 pandemic [5,59]. Therefore, understanding vaccine hesitancy locally and the influence of social and behavioural barriers are critical in this regard, given that outcomes can be influenced by local factors related to the given contexts, vaccinations, and the people involved [1,60]. Identifying the determinants of vaccine hesitancy among the hesitant group and then tailoring the vaccination campaign to fit this group is essential for behavioural change communication [2]. To address the problem, inclusive health communication centred on the empirical analysis of social and behavioural barriers is imperative to effective vaccination communication, which must be carefully planned and managed to encourage COVID-19 vaccine uptake [59]. Moreover, if poorly designed and executed, a COVID-19 vaccination campaign could undermine increasingly tenuous beliefs in vaccines and the public health authorities that recommend them [1,8].

This study employs TPB as a theoretical guide to develop an understanding of COVID-19 vaccine hesitancy behaviours from YouTube users’ comments. Although YouTube has greatly contributed to information sharing and receiving in public health, little attention has been paid to the factors that influence the intention of YouTube users towards COVID-19 vaccine hesitancy and the consequences of such behaviour in relation to social and behavioural factors. Therefore, the basic tenets of TPB guide the data analysis to identify how YouTube users communicate about COVID-19, specifically vaccine-related issues. Therefore, the following research questions are put forth to better understand the links between the COVID-19 vaccination campaign in Nigeria and YouTube use:

RQ1: What is the tone of YouTube video headlines/titles related to the COVID-19 vaccination campaign in Nigeria?

RQ2: What is the tone of Nigerian YouTube users’ comments on Nigerian COVID-19 vaccine videos?

RQ3: What causes of COVID-19 vaccine hesitancy are identified within Nigerian YouTube video user comments?

## 2. Methods

Videos uploaded between March 2021 and December 2022 were analysed. These dates were chosen because March 2021 was when the COVID-19 vaccination was launched in Nigeria, and the expected 70% threshold to fully vaccinate and eradicate the COVID-19 pandemic in Nigeria was expected to be reached by December 2022. A search for videos on www.YouTube.com (accessed on 6 December 2022) [61] with the keywords “COVID-19 in Nigeria”, “COVID-19 vaccination in Nigeria”, and “COVID-19 vaccine hesitancy in Nigeria” was conducted. The search produced 319 videos in English. The videos were further sorted by relevance, which is in line with previous studies [22,62]. The primary researcher observed that 65 videos did not meet the criteria for inclusion because the videos were duplicates or posted before March 2021. Thus, 65 videos were excluded from the analysis, leaving 254 videos.

### 2.1. Coding Scheme

Each video was downloaded, and the link was saved. The following two features were selected for coding to understand the tone of the videos: (1) the title/headline of the video and (2) all user comments. Titles/headlines and user comments were categorised as positive, negative, and/or neutral. Categorising media content and coding messages as positive, negative, or neutral/not applicable is a common practice in content analysis [22,62,63]. Following a similar practice, in this study, messages coded under a positive tone were classified as any information that conveys news about the existence of COVID-19 and supports COVID-19 safety protocols and vaccination(s). A negative tone was classified as any information that contests the existence of COVID-19 and disapproves of the vaccination. A neutral/not applicable tone was classified as any information that neither approves nor disapproves or is silent on the existence of COVID-19 and instead focused on information that is not applicable to the study context.

### 2.2. Coding

The primary researcher developed a codebook to explore the causes of COVID-19 vaccine hesitancy from the comments made by YouTube users. Most of the YouTube videos have turned off their comments section; nevertheless, 57 Nigerian mainstream media YouTube videos had their comments available; therefore, an online tool (exportcomments.com accessed 7 February 2023) [64] was used to generate comments from the 57 videos with accessible comments. A total of 985 comments were retrieved, downloaded and saved on Microsoft Excel for analysis. The list of themes for comments associated with vaccines on social media developed by Broniatowski et al. [65] was modified and utilised.

### 2.3. Intercoder Reliability/Agreement

Reliability within content analysis refers to its stability, which is the tendency for all assigned coders to consistently recode the same data in the same way over a period of time. Reproducibility is the tendency for a study group of coders to classify thematic categories in the same way [66]. To ensure the reliability of this study, two coders independently coded the data. According to Macnamara [67], for maximum reliability in media content analysis, it is required that two or more coders are used, at least for a sample of content (called the reliability subsample). The intercoder reliability for coding the headlines/titles was 76.77%. The intercoder reliability for coding users’ comments was 79.03%.

## 3. Results

To explore RQ1 about the tone of YouTube videos related to the COVID-19 vaccination campaign in Nigeria, all videos’ titles/headlines were coded. Of the 254 videos, the majority (53.5%) were positive in tone (*n* = 136), while 40.5% (*n* = 103) were negative, and the remaining 6% (*n* = 21) were neutral/not applicable.

To explore RQ2 about the tone of Nigerian YouTube users’ comments on the Nigerian COVID-19 vaccine, 985 comments were generated from 57 videos with accessible comments. The comments were coded (CM1–CM 57). The analysis shows 32.4% of the comments were negative (*n* = 319) and 5.0% of the comments were positive in tone (*n* = 49). However, 62.6% of the comments were neutral/not applicable to the analysis (*n* = 617). This can be attributed to two reasons. First, many of the comments generated were discussing travel, politics, etc. and did not discuss vaccination(s). Second, many comments were “antivaccine” emojis and did not include textual content.

To explore RQ3 about the causes of COVID-19 vaccine hesitancy identified within Nigerian YouTube video user comments, a thematic analysis was conducted following Broniatowski et al.’s [65] thematic approach. This analysis identified 319 negative and 49 positive comments among a total of 368 YouTube users’ comments. Table 1 presents antivaccine themes and provaccine themes, including the number of comments analysed under each category and exemplified comments.

## 4. Discussion

This paper examined the influence of social and behavioural barriers to the COVID-19 vaccination campaign in Nigeria. First, content analysis assessed the tone of YouTube video headlines/titles related to the COVID-19 vaccination campaign in Nigeria. Second, the study analysed Nigerian COVID-19 vaccine YouTube videos and described the tone of YouTube users’ comments about COVID-19 vaccines. Third, a thematic content analysis explained the causes of COVID-19 vaccine hesitancy identified within Nigerian YouTube video user comments.

Results of the findings show Nigerian COVID-19 vaccination campaign headlines/titles on YouTube from the government were predominantly positive (53.5%). In contrast, findings of YouTube users’ comments revealed that while 62.6% were neutral/not applicable in the analysis, the remaining 32.4% of comments were negative, while only 5.0% were positive in tone about the COVID-19 vaccination. These findings corroborate Briones et al. [22], who found public discussions on YouTube about the HPV vaccine were predominantly (51.7%) negative and suggested “future research could determine whether the source and tone of the video, as well as the specific content, are related to the negativity or positivity of the comments” (p. 485). The results of the current study confirm the tone of headlines/titles about COVID-19 vaccination messages from the government were positive; nevertheless, the tone of users’ comments on COVID-19 vaccines was largely negative. The present study also used thematic content analysis to explore the causes of vaccine hesitancy according to YouTube users’ comments. Most respondents who critiqued vaccines in their comments (*n* = 319) posted about how governments cannot be trusted on vaccines, as well as general vaccine conspiracy theories. Those who promoted vaccinations tended to comment on how vaccines work.

The results of this study provide some support for the explanatory nature of the Theory of Planned Behaviour. User comments about the government cannot be trusted on vaccines and freedom of choice/antimandatory vaccines often articulated that vaccines should not be required. These comments were prompted by social movements and human rights advocacy groups that challenged the government on mandatory vaccination. The theory of planned behaviour suggests people’s behaviour is determined by their attitudes, subjective norms, and perceived behavioural control [49]. In this regard, subjective norms suggest individuals (e.g., opinion leaders, social institutions, or pressure groups) shape the beliefs of most people who matter to them and believe they should or should not engage in the behaviour in question [49,50]. As such, their attitude towards the behaviour determines the degree to which a person(s) views an action as being favourable or unfavourable [51]. In this case, subjective norms shaped the user’s behavioural intentions and stimulated negative behavioural control towards taking the jabs as unfavourable towards their “freedom of choice”. On the other hand, the analysis also showed fear of vaccines due to safety-related concerns. People are hesitant of vaccines for fear of vaccine safety, mentioning the “odds of dying greater taking the vaccines” than being infecting with COVID. This clearly showed people are generally misinformed about COVID vaccinations. Krishna [39] confirms misinformation has created a barrier to people’s ability to attain health literacy since those armed with incorrect knowledge would be ill-equipped to make proper health decisions. According to the theory of planned behaviour, Ajzen, [52] predicts that behavioural beliefs can influence how people exhibit a positive or negative attitude towards the behaviour. Therefore, Breslin et al. [59] confirm some people will only agree to take the COVID-19 vaccine if there are no reported side effects. In Africa in particular, Limbu et al. [60] discovered one of the main factors driving vaccination intentions was perceived behavioural control.

In addition, study found that users’ comments on general conspiracy-related theories dominated discussions online. Our findings provided further support that misinformation on social media is a threat to vaccination campaigns, especially, in the COVID-19 vaccination campaign. Montagni et al. [35] observed that since the beginning of the COVID-19 pandemic, fake news has been widely disseminated, making people distrust the integrity of information from healthcare services and political authorities. In Nigeria, Abayomi [68] confirms the lack of openness and accountability in the COVID-19 pandemic’s response has exacerbated the deteriorating public trust. The findings also revealed expressions of religious beliefs and biotechnology-related conspiracies dominated users’ discussions about COVID-19 vaccines, for instance, some users believe, “Take the vaccine, you die or lose your fertility”, while others suggest “It’s mRNA Technology…”, and most shockingly, some said “Jesus is my vaccine” or the vaccine is a “Mark of the beast”. This type of religious misinformation and conspiracy theory was mentioned, for example, in Makurdi, Nigeria, where vaccine hesitancy has persisted due to aphorisms about Jewish expectations of the anti-Christ, such as the mark of the beast, being chained to hell, acting immorally, and having animalistic inclinations, among others, according to research by Uroko and Okuosa [69].

## 5. Implications for Theory

### 5.1. Misinformation and Trust

Although TPB-guided data analysis in this study, it has not adequately addressed the underlying beliefs that shape the attitudes of Nigerians towards COVID-19 vaccine hesitancy. Generally, from this analysis, we fully understand that sharing misinformation unabated during a pandemic can have greater consequences that could possibly cause a lack of trust in messages emanating from public authorities. Lovari [70] observed that disseminating misinformation, especially during a pandemic like COVID-19, if not properly managed, can amplify risk behaviours that could potentially be harmful to the people. Consequently, vaccine hesitancy was likely due to misinformation and a lack of trust in the Nigerian government’s messages. The unrelenting misinformation from sources other than government institutions led to a lack of trust; thus, the overriding concepts in public safety campaign messages have mostly been ignored by many Nigerians. In this case, TPB could not fully explain vaccine hesitancy among Nigerians in relation to what and how the sources of misinformation influence distrust of the government’s messages to be negatively perceived.

### 5.2. Framing

The second implication of the study lies in the inherent framing of headlines/titles. Although mostly positive, the choice of words in framing the headlines/titles mostly contradicts the intended meaning. For example, the headlines/titles “Did people’s power really lead to COVID-19 vaccines being destroyed in Nigeria?”, and “Around one million doses of COVID-19 vaccines wasted in Africa’s Nigeria” were considered negative in this analysis. The unsuitable choice of words can lead to a lack of understanding about the importance of vaccinations; hence the desired meaning of the message was not effectively communicated. Philosophically, frames are meant to select given aspects of identified realities and make them salient to draw positive public attention and gain support. Therefore, YouTube users might be potentially misinformed about the importance of the government’s COVID-19 vaccination messages due to the incorrect choice of frames in some of the headlines/titles analysed.

### 5.3. Practical Implications and Policy Contributions

Content creators were part of the provaccine themes that effectively influenced vaccine messages by educating people about vaccines protecting herd immunity as exemplified in this comment, “This is great content so that people will be aware of the importance of vaccination”. This study, therefore, proposes that governments should adopt collaborative strategies with trained content creators to develop proactive risk communication, with a top-down and bottom-up communication approach, specifically, with local healthcare providers, mass media, community/traditional chiefs, religious leaders or organisations, and pressure groups to ensure people are well-informed and adequately enlightened about the benefits of vaccines. Most importantly, due to inadequate public amenities in many parts of Nigerian communities, a door-to-door COVID-19 vaccination awareness campaign should be implemented. In a recent study, Adebimpe and Adeoye [71] suggested one significant method being utilised to improve vaccination at the state level is door-to-door vaccination.

Furthermore, the findings of this study highlighted the paucity of public trust, ineffective public health policies, and lack of accountability in the government of Nigeria, which cause vaccine hesitancy. The findings of this study will help governments in designing communication strategies to promote public trust in vaccines and increase public access to verified information based on the safety and efficacy of vaccines. This can be achieved through messages logically designed to increase public trust on the basis of honesty and transparency, partly providing sufficient information on the vaccine production process, ingredients, effective administration, and how to combat counterfeit vaccines. Sato [72] suggests when people had more trust in the government, it is more likely for them to accept the COVID-19 vaccine, since vaccine hesitancy in Nigeria is partly related to people’s distrust of the government. Second, the study can guide policymakers to provide evidence-based campaign strategies on the safety and effectiveness of vaccines; thus, practically outlining the benefits of vaccines and communicating effectively to reduce barriers of perceived risks, removing such risk perceptions, should be the core of the vaccination campaign in fostering public confidence [73]. Third, these findings can guide public policy directions on accountability. Policymakers will understand that being accountable to the people during pandemics through socioeconomic measures will ameliorate people’s suffering and hardships and consequently help address the challenges of vaccine hesitancy and increase vaccine uptake in Nigeria.

In summary, social and behavioural barriers will continue to present obstacles to Nigeria’s ambition to end the scourge of the COVID-19 pandemic due to the presence of vaccine hesitancy; hence, this will also affect Nigeria’s national vaccination plans. Currently, the WHO database shows that only 33.86% of people are fully vaccinated against COVID-19 in Nigeria, as of 22 May 2023 [74]. The vaccine uptake is very low considering the massive population of Nigeria of approximately 216 million people. Therefore, to increase the ratio of vaccine uptake, as a suggestion for policy making, the government must develop the appropriate collaborative interventions, where all the critical stakeholders (i.e., opinion leaders, traditional/cultural and religious leaders) should be involved in the vaccination education campaigns to bridge the lack of trust in the government and debunk misinformation and fears about the safety and efficacy of COVID-19 vaccines.

## 6. Limitations and Future Research

The study is limited to the analysis of YouTube videos; however, future research should investigate the impact of misinformation and trust in COVID-19 vaccination communication expanded from YouTube to Facebook and Twitter, as figures reveal that around 33 million Nigerians utilise these platforms. Additionally, the current political landscape was saturated with 2023 political and electioneering campaign activities; consequently, social distancing protocols were ignored. This may have increased the number of COVID-19 cases in Nigeria. Therefore, based on this observation, future research should examine the role of COVID-19 vaccination misinformation and trust in Nigeria’s 2023 general election campaign.

## 7. Conclusions

From the findings of this study, three key conclusions were made. Based on the idea that misinformation has caused a lack of trust in the Nigerian government’s COVID-19 vaccination messages, this research has proposed the following three key areas and solution strategies to address misinformation about vaccination campaigns in Nigeria.

### 7.1. Administrative and Policy Directions

Given the negative impact of misinformation in public health campaigns, this study suggests policymakers at all levels of governance in Nigeria should promote health literacy campaigns as one of several critical strategies for reducing misinformation in public health. Particularly, media literacy should be encouraged, and increased training of information officers is required to address mediocrity gaps as part of proactive pre- and postpandemic communication management. This means providing instructional materials to help people recognise and evaluate the accuracy of information as well as the credibility of various online information outlets is critical. The aim of media literacy is to shape people’s ability to obtain, process, and understand basic health information and services necessary for making informed health decisions, as Sørensen et al. [47] found that negative media misconceptions of vaccines can decrease the demand for vaccinations and increase vaccine hesitancy [32].

### 7.2. Development of Healthcare Professionals

In Nigeria, healthcare professionals are lagging in terms of training. Frontline healthcare workers in particular should be trained and equipped with effective communication skills needed to effectively communicate safer paths during public health emergencies like the COVID-19 pandemic. Shaw [75] observed many patients have little knowledge of medical terminologies; consequently “layman’s” language is paramount to ensure that messages are accurately conveyed while conversing with people.

### 7.3. Research Collaborations

To fully understand the root causes and consequences of public health misinformation, this study proposes that governments at all levels in Nigeria, including local and international healthcare organisations, should encourage research and development. Findings of research like the ongoing study will help determine the sources of misinformation and suggest effective measures to address such problems.

## Figures and Tables

**Table 1 vaccines-11-01057-t001:** Examples of YouTube Users’ Comments on the COVID-19 Vaccination and Corresponding Themes: March 2020–December 2022.

Theme (A)	Antivaccine Themes	Comments (*n*)	Example Comments
Theme 1	Freedom of choice/antimandatory vaccines	12.22% (*n* = 39)	“Free and informed medical consent”. “My body, my choice! not going to be bullied, coerced, or threatened into taking something the Nigerian government or scientists don’t have a clue about its contents”. “Not one Nigerian scientist can tell us what is contained in that vaccine it’s telling people to inject into their veins”. “People should be allowed to make decisions for their health. COVID-19 is a scam”. “Making it compulsory will only increase the suspicion index of citizens concerning the vaccine…wrong advice for real”. “Stand for FREE CHOICE”.
Theme 2	Cannot trust government on vaccines	15.7% (*n* = 50)	“Sell-out leaders as always. Know shame. Be wise and don’t even accept this from them”. “Can the Nigerian Government be trusted? Those who received the stimulus money should be vaccinated”. “The manufacturers don’t trust their vaccine and you have to sign an indemnity”. “And you administer this to humans”. “If you said the vaccine is safe and that people with the vaccine are safe from contracting the virus or spreading the virus, why is the government so desperate of imposing the vaccine if they are not puppets to those, they’ve collected funds from”. “Nigeria leaders ready to sacrifice Nigerians in order to please Bill Gates”.
Theme 3	Pharmaceutical companies want vaccine profits	1.88% (*n* = 6)	“Viruses are a product of the Earth’s ecological cycles. They regulate the planet’s ecosystems and evolve life forms. The Earth also produces Humans, and our populations are also regulated by the planet in order to maintain our ecosystems”. “Vaccines are not naturally occurring they don’t adapt and change like our body’s immune system and some people only care about getting rich”. “COVID-19 is a business”
Theme 4	Vaccines cause bad side effects	12.53% (*n* = 40)	“Your odds of dying are greater getting the vaccine than getting COVID”. “Are you guys, okay? the manufacturer says they are not liable if anything happens”. “This is absolutely rubbish, common sense is not common at all”. “Don’t COMPLAIN if some of them DIE or their health is AFFECTED because of the VACCINES. Many symptoms occur after the vaccine is applied”. “Similar side effects with elephantiasis vaccine”
Theme 5	General vaccine conspiracy theories	46.08% (*n* = 147)	“My people die because of the lack of knowledge. Lucifer and his messengers of the new world order are on rampage, making sure the entire world becomes his prey! As for me and my household, we will not bow to the gods of Babylon, which is what this mass vaccination campaign is all about! This is the most sophisticated crusade humanity has ever seen! It’s so sad Christians can’t see the hand of Satan in all this!” “Stop! They’re taking pictures like it’s a spectacle! Jesus is my vaccine. Please repent” “If you take the vàcx, forget it, you’re going straight to hell, for you have taken the mark of the beast… you become a GMO”. “Take the vaccine, you die or lose your fertility”. “It’s mRNAa TECHNOLOGY. their words”. “It is not medicine. Elon musk: “mRNA technology is like a computer program in your body”.
Theme 6	Vaccine ingredients are dangerous	9.71% (*n* = 31)	“This YouTuber is a fool for encouraging people to take the poisonous vaccines”. “No, to COVID-19 poisonous vaccines”. “Not realizing 💉 is just poison, folk in the US”. “Nigerians should know better! #saynotothepoisonjab #thejabislethal”
Theme 7	Diseases are not so dangerous	1.88% (*n* = 6)	“Why take a shot when it goes away on its own wake-up”. “Take off the masks Trust Me it’s not needed. Peace”. “Hard to listen or take serious people who wear mask”. “Funny that my friend living in Bakomo Africa said nobody wears masks and nobody has it…”.
	Total	100%	
Theme (B)	Provaccine themes		
Theme 8	Vaccines work	61.2% (*n* = 30)	“Funky made it… thanks for sharing your experience. I have taken my first (Pfizer) vaccine dose here in the US (North Carolina). The only side effect I can think of was a pain in my arm where the injection took place, it only lasted overnight. The next day was fine”. “Am fully vaccinated and boosted”. “Thank God the vaccine is now in Nigeria……can’t just wait”.
Theme 9	Vaccination protects herd immunity	12.2% (*n* = 6)	“This is great content so that people will be aware of the importance of vaccination”. “Thank you for encouraging everyone to get vaccinated to protect themselves from COVID-19”.
Theme 10	People who do not vaccinate put me/my kids at risk	2.0% (*n* = 1)	“Origbo, Haba can’t we for once celebrate good news? Please can you share with us why unvaccinated people should be allowed into 9ja? I just returned to the United States from 9ja last week and did COVID-19 test a day before departure. It’s a requirement by the American government. What’s wrong with requiring unvaccinated to get it done? Thank you!”
Theme 11	Alternative medicine does not work	24.6% (*n* = 12)	“Actually, not advised to take ibuprofen due to the side effect of possible bleeding. Note!! I said possible not compulsory. Safest is paracetamol. One of the screening questions we ask here in the UK is “do you have a bleeding disorder and also do you take anticoagulants. therefore, ibuprofen is not advised”.
	Total	100%	

## Data Availability

Data will be made available upon reasonable request to the corresponding author.

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
