# Peer review of "COVID-19 Vaccine Hesitancy: A Content Analysis of Nigerian YouTube Videos"

_vaccines, 2023, doi:10.3390/vaccines11061057_

Round 1
Reviewer 1 Report
Thank you for the invitation to review this manuscript. The manuscript has sound methodology. However, I have a few concerns that needed to be addressed.
1. I am not sure that why authors from New Zealand are particularly assessing the objectives? Is there any author native to Nigeria? How can authors from other countries discuss the political, social and economic conditions of a country where they are not residing?
2. The conclusions should provide major findings along with directions for future researchers, policy makers and the healthcare system.
3. Please provide the limitations section before conclusion.
4. There is a need to provide current vaccination situation in Nigeria in the discussion section.
5. Vaccine hesitancy is a global phenomenon. How it differs in Nigeria? If it follows the same pattern as reported by many studies during the COVID-19 pandemic, then what is the rational of this study?
6. Misinformation should be addressed through appropriate policies and directives from Government rather than future research, please clearly describe the solutions of each problem starting from administration to healthcare professionals to researchers.
Author Response
Reviewer’s comments |
Response |
Reviewer 1 · I am not sure that why authors from New Zealand are particularly assessing the objectives? Is there any author native to Nigeria? How can authors from other countries discuss the political, social and economic conditions of a country where they are not residing? |
Thank you for the comment. The primary author is Nigerian. |
· The conclusions should provide major findings along with directions for future researchers, policymakers and the healthcare system. |
We have added a section to the paper on pages 15-16 for policy contributions.
|
· Please provide the limitations section before conclusion. |
The paragraph is moved up under the subheading> Limitations (pg. 16). |
· There is a need to provide current vaccination situation in Nigeria in the discussion section. |
More has been added on pages 4 and 16. |
· Vaccine hesitancy is a global phenomenon. How it differs in Nigeria? If it follows the same pattern as reported by many studies during the COVID-19 pandemic, then what is the rational of this study? |
Added on page 4.
|
· Misinformation should be addressed through appropriate policies and directives from Government rather than future research, please clearly describe the solutions of each problem starting from administration to healthcare professionals to researchers. |
More has been added to this point on pages 4 and 17-18
|
Reviewer 2 Report
Thank for giving me the opportunity to review this interesting manuscript. Authors analyzed the Nigerian YouTube videos and their comments section to understand causes of COVID-19 vaccine hesitancy in Nigeria. The manuscript is well-written, for the most part, and adheres to the journal’s guidelines but the description at several points throughout the paper are lacking which needs to be addressed by the authors.
Abstract:
1. Describe the materials and methods used in this study.
2. “Results show the tone in (n = 254) of …… 6% was neutral”. Rephrase this sentence to make it clearer and comprehensible for the readers.
3. “Second, findings indicate the majority of the Nigerian YouTube users’ comments were neutral 62.6% were neutral, while 32.4, were negative and 5% were positive”. Please correct this sentence by removing the repetitions. Kindly place “62.6%” in the parenthesis. Also, insert “%” symbol after 32.4.
4. Briefly summarize your study's conclusions, implications for research and the practice in the final sentence of your abstract.
Introduction: This section is unnecessarily long. There is some repetitive information e.g. details on the theory of planned behavior are given at two separate places). Kindly revise and shorten the introduction section.
Methods
Ø Were there any Nigerian YouTube videos that weren’t in English? If yes, why those were not considered for analysis?
Ø If possible, add a study flow chart/diagram.
Results
Ø How many of these 57 videos had positive, negative and/or neutral tone?
Ø No need to mention that you used exportcomments.com to gather the comments as it has already been mentioned in the methods section.
Ø Better to exclude Theme 5, 7, 11, 12, 15, 16, 18 from the Table 1 as there was no content that fits these themes (n = 0).
Author Response
Reviewer’s comments |
Response (please note, my responses have also been highlighted in Yellow in the main document). |
· Reviewer 2 |
|
· Describe the materials and methods used in this study. |
Edited |
· “Results show the tone in (n = 254) of …… 6% was neutral”. Rephrase this sentence to make it clearer and comprehensible for the readers. |
Edited |
· “Second, findings indicate the majority of the Nigerian YouTube users’ comments were neutral 62.6% were neutral, while 32.4, were negative and 5% were positive”. Please correct this sentence by removing the repetitions. Kindly place “62.6%” in the parenthesis. Also, insert “%” symbol after 32.4. |
Edited |
· Briefly summarize your study's conclusions, implications for research and the practice in the final sentence of your abstract. |
Edited |
· Introduction: This section is unnecessarily long. There is some repetitive information e.g. details on the theory of planned behavior are given at two separate places). Kindly revise and shorten the introduction section. |
We have edited the introduction and moved the discussion of TPB to the review. |
Were there any Nigerian YouTube videos that weren’t in English? If yes, why those were not considered for analysis? |
This has been clarified on page 9. |
· If possible, add a study flow chart/diagram. |
We have considered this recommendation, and while an interesting idea we have decided not to add this diagram at this time. |
· How many of these 57 videos had positive, negative and/or neutral tone? |
Overall, the 57 videos, as coded CM1-CM57, had mixed of positive, negative, and neutral comments across the board (e.g., CM 6 has only 4 negative comments, CM 14 has 5 negative and 7 neutral comments, CM 18 has 8 neutral comments, and CM 22 has 139 neutral, 3 negative and 1 positive comment, CM 48 has 13 negatives and 3 neutral comments. While CM 57 has only 1 negative comment). |
· No need to mention that you used exportcomments.com to gather the comments as it has already been mentioned in the methods section. |
Edited |
· Better to exclude Theme 5, 7, 11, 12, 15, 16, 18 from the Table 1 as there was no content that fits these themes (n = 0). |
Themes 5, 7, 11, 12, 15, 16, & 18 have been excluded and table edited. |
Round 2
Reviewer 2 Report
Thank you, the manuscript has been improved.